# In Situ Vaccination by Tumor Ablation: Principles and Prospects for Systemic Antitumor Immunity

**DOI:** 10.3390/vaccines13111114

**Published:** 2025-10-30

**Authors:** Tinatin Chikovani, Eli Magen

**Affiliations:** 1Department of Immunology, Tbilisi State Medical University (TSMU), Tbilisi 0186, Georgia; t.chikovani@tsmu.edu; 2Department of Internal Medicine A, Assuta Ashdod University Medical Center, Faculty of Health Sciences, Ben-Gurion University of the Negev, Beer Sheva 8410501, Israel

**Keywords:** in situ vaccination, tumor ablation, immunogenic cell death, damage-associated molecular patterns, abscopal effect, tertiary lymphoid structures, immune checkpoint inhibitors, cGAS-STING pathway, biomarkers, combination immunotherapy

## Abstract

Cancer immunotherapy has redefined oncology’s goals, aiming for durable systemic immunity rather than mere cytoreduction. However, many solid tumors remain refractory due to immunosuppressive microenvironments and antigenic heterogeneity. Local tumor ablation techniques—including radiofrequency ablation (RFA), microwave ablation (MWA), cryoablation, irreversible electroporation (IRE), and high-intensity focused ultrasound (HIFU)—are being re-evaluated beyond their historic cytoreductive role. This comprehensive review synthesizes the paradigm of tumor ablation as an in situ vaccination strategy, a concept that leverages the tumor itself as a source of antigens and the ablation process to generate endogenous adjuvants. We detail the mechanistic underpinnings, highlighting how ablation induces immunogenic cell death (ICD), releasing damage-associated molecular patterns (DAMPs) such as calreticulin, ATP, HMGB1, and cytosolic DNA. These signals activate innate immunity via pathways like cGAS-STING, promote dendritic cell maturation, and facilitate epitope spreading. We critically examine the determinants of efficacy, including the critical impact of ablation modality on the “DAMP signature,” the necessity of complete ablation, and the pivotal role of the host’s immune contexture. Furthermore, we explore the induction of tertiary lymphoid structures (TLS) as a key anatomical site for sustained immune priming. Translational strategies are extensively discussed, focusing on optimizing procedural techniques, rationally combining ablation with immune checkpoint inhibitors (ICIs) and innate immune agonists, and developing a robust biomarker framework. By adopting the core principles of vaccinology—meticulous attention to antigen, adjuvant, route, and schedule—ablation can be engineered into a reproducible platform for systemic immunotherapy. This review concludes by addressing current limitations and outlining a roadmap for clinical translation, positioning interventional oncology as a central discipline in the future of immuno-oncology.

## 1. Introduction

The landscape of oncology has been fundamentally reshaped by immunotherapy, shifting the therapeutic paradigm from direct tumor cytotoxicity to empowering the host’s immune system to achieve durable, systemic disease control. Immune checkpoint inhibitors (ICIs), which target regulatory pathways such as PD-1/PD-L1 and CTLA-4, have demonstrated unprecedented long-term survival benefits across a spectrum of malignancies [1,2]. Similarly, bispecific T-cell engagers and adoptive cell therapies, such as CAR-T cells, represent a new frontier of engineered immunity [3]. Despite these advances, a significant proportion of patients with solid tumors derive little or no benefit. Primary and acquired resistance remain formidable challenges, often driven by an immunosuppressive tumor microenvironment (TME), low tumor mutational burden (TMB), poor antigen presentation, and profound antigenic heterogeneity that allows for immune escape [4,5].

Historically, local tumor ablation techniques were developed as minimally invasive tools for cytoreduction. Modalities such as radiofrequency ablation (RFA), microwave ablation (MWA), cryoablation, irreversible electroporation (IRE), and high-intensity focused ultrasound (HIFU) offered a means to destroy tumors percutaneously with minimal morbidity, particularly for patients who were poor surgical candidates [6,7]. The primary endpoint was local control, with any systemic effect considered an ancillary benefit, if considered at all [7].

Over the past decade, this perspective has undergone a radical transformation. A growing body of evidence suggests that the therapeutic effect of ablation extends far beyond the physical destruction of tissue [8,9]. It is now understood that ablating a tumor in situ can initiate a cascade of immunologic events that closely mimic a conventional vaccination. This has given rise to the powerful concept of “in situ vaccination” [10].

The in situ vaccination paradigm posits that the controlled destruction of tumor tissue in its native anatomical and immunological context provides the two fundamental components of a classical vaccine:**Antigen Source**: The tumor itself serves as a comprehensive, personalized library of tumor-associated antigens (TAAs) and neoantigens. This library reflects the full clonal heterogeneity of the malignancy, presenting a diverse antigenic repertoire to the immune system that is impossible to replicate with exogenously manufactured vaccines fully [11].**Endogenous Adjuvant**: The ablation procedure induces a specific form of cell death known as immunogenic cell death (ICD). ICD is characterized by the spatiotemporal release of endogenous danger signals, or damage-associated molecular patterns (DAMPs) [12]. These include pre-apoptotic surface exposure of calreticulin (an “eat-me” signal), extracellular release of ATP and HMGB1, and the leakage of cytosolic DNA [13,14]. These DAMPs bind to pattern recognition receptors (PRRs) on innate immune cells, activating critical pathways such as cGAS-STING and triggering robust type I interferon (IFN-I) responses, which are essential for effective dendritic cell (DC) licensing and T-cell priming [15,16,17].

From a vaccinology perspective, ablation thus serves a sophisticated function: it transforms the tumor into an in situ depot that co-delivers a personalized antigen payload with a potent, locally released adjuvant cocktail [18,19]. In clinical oncology, the most striking manifestation of this systemic immunity is the abscopal effect (from the Latin ab—“away from” and scopus—“target”)—the regression of metastatic lesions outside the localized treatment field following local therapy [19]. This conceptual reorientation aligns with the broader trend in oncology of re-interpreting local physical interventions—most notably radiotherapy—in immunologic terms. The abscopal effect, a phenomenon in which localized treatment induces regression of metastatic disease at distant sites, is the classic manifestation of systemic immune activation [20,21,22,23].

The unique appeal of ablation as an in situ vaccine lies in its universality and simplicity. Unlike peptide, DNA, or dendritic-cell vaccines that require prior antigen discovery, prediction, synthesis, and complex ex vivo manufacturing, ablation is agnostic to tumor genotype and histology [24,25]. Each ablated lesion automatically provides the patient’s complete, native antigenic repertoire in its natural conformation and post-translational state, presented within the powerful inflammatory context of tissue disruption and DAMP release [24,26]. Consequently, ablation-induced immunity has the potential to overcome the significant challenges posed by inter- and intra-tumoral heterogeneity, a major limitation of current targeted immunotherapies [27]. However, the efficacy of this approach can be modulated by tumor-intrinsic factors, such as the suppression of key innate immune sensing pathways, which may predict resistance and require additional combinatorial strategies [28,29,30]. Table 1 compares classic systemic vaccination with in situ vaccination via tumor ablation, highlighting their similarities, differences, advantages, and disadvantages.

In this extensive review, we delve into the mechanistic foundations underlying tumor ablation as an in situ vaccination platform. We critically examine the key determinants of its efficacy, drawing explicit parallels to classical vaccinology principles of antigen, adjuvant, route, and schedule. We then propose and elaborate on advanced translational strategies to standardize and enhance this approach, focusing on combination therapies with ICIs and innate immune agonists, as well as on the development of a rigorous biomarker framework. Finally, we survey the evolving clinical trial landscape, discuss safety considerations, address current limitations, and offer a forward-looking perspective on integrating this promising modality into the next generation of cancer immunotherapy.

## 2. Mechanistic Basis of Ablation as Vaccination

### 2.1. Antigenic Breadth and the Catalyst of Epitope Spreading

A fundamental tenet of successful vaccination is the exposure of the immune system to a sufficient breadth of antigens to stimulate a diverse, durable, and robust immune response capable of overcoming escape variants. Tumor ablation achieves this organically by simultaneously releasing a vast array of canonical tumor-associated antigens (e.g., MUC1, CEA, HER2, NY-ESO-1) alongside a unique set of patient-specific neoantigens generated by the tumor’s individual mutational landscape [26,31]. This stands in stark contrast to synthetic peptide or mRNA vaccines, which are inherently restricted to a limited set of predefined epitopes, often selected based on bioinformatic predictions that may not accurately reflect actual immunogenicity or clonal representation [32,33].

This comprehensive antigenic exposure is the critical catalyst for a process known as epitope spreading (also called antigen spreading or determinant spreading) [34]. This is a phenomenon in which the immune response, initially directed against a limited number of immunodominant epitopes released by the ablation, expands over time to recognize additional, secondary antigens present on tumor cells [35]. This occurs through the efficient phagocytosis of apoptotic/necrotic tumor debris by antigen-presenting cells (APCs), which process and present a much wider array of antigens than were initially targeted [36,37]. Epitope spreading is a hallmark of a productive, broadening immune response and a key defense against tumor escape by downregulating single antigens [34,38].

In preclinical models, ablation-induced epitope spreading correlates directly with the development of a more diverse T-cell receptor (TCR) repertoire intratumorally and peripherally, and is a strong predictor of improved local and systemic tumor control [39]. Clinically, epitope spreading has been identified as a key correlate of response and durable clinical benefit in patients treated with ICIs and other immunotherapies [40]. It may account for the long-term, ongoing regression seen in a subset of patients. Thus, the value of ablation lies not only in the quantitative release of a large antigenic payload but also in its qualitative ability to drive the diversification of immune recognition, thereby significantly reducing the likelihood of clonal escape and tumor recurrence [41].

### 2.2. Immunogenic Cell Death: The Engine of Endogenous Adjuvanticity

The immunologic potency of ablation is not merely a function of antigen release; it is critically dependent on the quality of cell death [42]. Immunogenic cell death (ICD) is a functionally unique form of cell death that activates the adaptive immune system against dead-cell-associated antigens. It is characterized by the stereotyped, sequential emission of stress and damage signals that, together, act as a powerful endogenous adjuvant system [43,44].

The well-defined biochemical hallmarks of ICD include:Pre-apoptotic exposure of calreticulin (CRT): Normally residing in the endoplasmic reticulum, CRT translocates to the cell surface during early ICD. It acts as a potent “eat-me” signal, engaging CD91 receptors on phagocytes such as dendritic cells and macrophages, thereby promoting the phagocytosis of tumor cell corpses and subsequent cross-presentation of tumor antigens to CD8^+^ T cells [43,45].Extracellular release of adenosine triphosphate (ATP): Released from dying cells, ATP serves a dual function. First, it acts as a potent chemoattractant for monocytes and dendritic cells by engaging P2Y2 receptors [46,47]. Second, it binds to the P2RX7 purinergic receptor on the surface of dendritic cells, triggering the assembly and activation of the NLRP3 inflammasome [43,48]. This leads to the secretion of pro-inflammatory cytokines, such as IL-1β, which is critical for the polarization of IFN-γ-producing CD8^+^ T cells [49,50].Release of high-mobility group box 1 (HMGB1): This nuclear protein is released passively during late-stage necrosis. HMGB1 binds to Toll-like receptor 4 (TLR4) on dendritic cells, promoting their maturation, enhancing antigen processing, and stimulating the production of pro-inflammatory cytokines such as TNF-α and IL-6 [51]. The interaction between HMGB1 and TLR4 is crucial for effective cross-priming [51].Leakage of cytosolic and nuclear DNA: The loss of membrane integrity leads to the release of DNA into the extracellular space and the TME [52,53,54]. This DNA is sensed by the cyclic GMP-AMP synthase (cGAS), which catalyzes the production of cyclic GMP-AMP (cGAMP) [55]. cGAMP acts as a second messenger that activates the stimulator of interferon genes (STING) pathway in surrounding dendritic cells and other immune cells [56]. STING activation is a master regulator of type I interferon (IFN-I) production, arguably the most critical cytokine linking innate immune sensing to adaptive T-cell immunity [57].

Collectively, this coordinated cascade of DAMPs licenses conventional type 1 dendritic cells (cDC1s)—a specialized subset equipped for cross-presentation—to efficiently phagocytose tumor debris, process the released antigens, and present them on MHC-I molecules to prime naive CD8^+^ T cells in the draining lymph nodes [58]. In the language of vaccinology, the ablation procedure provides a built-in, multi-component adjuvant system, effectively obviating the need for the synthetic adjuvants (e.g., alum, MF59, AS01) required in classical vaccine formulations [59]. It is important to emphasize that different ablation modalities yield distinct “DAMP signatures”—varying in the relative amounts and timing of these signals—which profoundly influence the magnitude, quality, and durability of the downstream adaptive immune response [26].

### 2.3. Anatomical Re-Engineering: Lymph Nodes and Tertiary Lymphoid Structures as Priming Hubs

Following ablation, the released antigens and DAMPs are transported via lymphatic drainage to the tumor-draining lymph nodes (TDLNs), which are the classical and primary sites for the initiation of adaptive immune responses [60]. Here, resident and newly arrived dendritic cells present tumor antigens to naïve T cells, leading to their activation, clonal expansion, and differentiation into effector T cells [61].

However, a more recently appreciated, and potentially equally important, mechanism is ablation’s ability to remodel the local tumor microenvironment to support the de novo formation and maturation of tertiary lymphoid structures (TLSs) [62]. TLSs are ectopic lymphoid aggregates that develop in non-lymphoid organs at sites of chronic inflammation, including cancer [63]. They are highly organized structures that recapitulate the architecture and functionality of secondary lymphoid organs, containing distinct T-cell zones, B-cell follicles, germinal centers, follicular dendritic cell networks, and high endothelial venules (HEVs) that facilitate lymphocyte entry [64].

The presence of mature, well-organized TLS within the TME is consistently associated with a favorable prognosis and improved responses to ICIs across multiple cancer types, including melanoma, lung cancer, and sarcoma [65,66,67,68,69]. They are thought to function as local “immune hubs” or “booster stations,” enabling ongoing antigen presentation, T-cell priming, B-cell activation, and antibody class switching directly at the tumor site [65,66,67,68,69,70]. This creates a sustained inflammatory microenvironment that can counteract local immunosuppression [70,71].

Spatial transcriptomics and single-cell immune profiling have revealed that TLS-rich tumors harbor expanded clones of T and B cells and show enriched signatures for IFN-γ signaling and germinal center reactions [72,73]. The post-ablation induction of TLSs can therefore function as a persistent local site for immune education and amplification, fostering durable T-cell/B-cell collaboration and maintaining immune pressure on the tumor [72,74]. Recent technological advances in spatial immune repertoire mapping, such as Slide-TCR-seq and CODEX, provide powerful tools to assess TLS maturation after ablation quantitatively and to track the spatial dynamics of epitope spreading and clonal expansion within these structures [75,76].

### 2.4. Systemic Manifestation: The Abscopal Effect and Beyond

Preclinical evidence robustly demonstrates that ablation can generate a wave of tumor-specific CD8^+^ T cells that enter the circulation, traffic to distant, non-treated metastatic lesions, and mediate their regression [77,78]. The defining feature of any successful vaccine is the establishment of systemic immunity that can survey and protect the entire organism. The immune basis of this effect is confirmed by experiments where depletion of CD8^+^ T cells, but not CD4^+^ T cells, completely abrogates the anti-tumor response at secondary sites [79,80].

While initially and most famously described in patients receiving radiotherapy, bona fide abscopal-like responses have now been reported following RFA, cryoablation, and IRE, particularly when these modalities are combined with ICIs [81]. Although the incidence of dramatic abscopal effects remains relatively low in retrospective analyses, these case reports and small series provide crucial proof-of-principle that ablation can indeed function as an effective in situ vaccine in humans [82]. The paramount challenge for the field is to transform these serendipitous observations into predictable and reproducible therapeutic outcomes through rational combination strategies, optimized treatment parameters, and patient selection based on biomarkers, including emerging predictors of innate immune resistance.

## 3. Determinants of Vaccinal Efficacy: A Vaccinologist’s Perspective

Although tumor ablation holds clear theoretical potential as an in situ vaccine, its immunologic outcome is not guaranteed and is highly context-dependent [83]. Not every ablation procedure results in productive systemic immunity; in some cases, it may even reinforce tolerance or accelerate immunosuppression by releasing inhibitory factors [10,84]. As with classical vaccinology, efficacy depends on the careful optimization of multiple interacting parameters [84].

### 3.1. Adjuvant Quality: Decoding the “DAMP Signature”

In vaccinology, the choice of adjuvant is paramount and tailored to induce a specific immune response (e.g., Th1 vs. Th2) [85,86]. Similarly, tumor ablation generates endogenous adjuvants in the form of DAMPs. Still, the profile, or “signature,” of these signals varies dramatically depending on the physical modality used and its specific technical parameters [87].

**Thermal Ablation** (RFA/MWA): These techniques induce rapid coagulative necrosis through high-temperature heating, resulting in the release of large quantities of tumor antigens [88]. However, extreme heat can also cause widespread protein denaturation, potentially destroying conformational epitopes and reducing antigenic fidelity [89]. Thermal ablation produces abundant ATP, HMGB1, and reactive oxygen species (ROS), which are potently inflammatory [90]. A significant limitation is the frequent creation of a perfused, sublethal peripheral rim due to heat-sink effects from adjacent blood vessels [91]. This rim of stressed but viable tissue often becomes a potent source of immunosuppressive cytokines, such as IL-6, VEGF, and HGF, which promote angiogenesis and recruit myeloid-derived suppressor cells (MDSCs) and regulatory T cells (Tregs) [92,93]. Thus, while thermal ablation generates strong adjuvanticity, it carries a concomitant risk of inducing counter-regulatory immunosuppressive mechanisms [94].**Cryoablation**: This modality utilizes freeze–thaw cycles to cause necrotic cell death. A key advantage is that it preserves the structural integrity of proteins and lipids, thereby maintaining antigenic fidelity and enabling T-cell receptors (TCRs) and B-cell receptors (BCRs) to recognize conformational determinants [95]. The necrotic death mode triggers a robust inflammatory response characterized by dense neutrophilic infiltration and the release of pro-inflammatory cytokines such as TNF-α and IL-1β. Clinical studies in breast and prostate cancer have shown that cryoablation is particularly effective at generating measurable systemic T-cell responses, especially when combined with checkpoint blockade [96]. A rare but serious risk is cryoshock, a systemic inflammatory response syndrome characterized by coagulopathy and multi-organ failure [97].**Irreversible Electroporation (IRE)**: A non-thermal modality that uses high-voltage electrical pulses to create permanent nanoscale defects in cell membranes, leading to apoptosis and secondary necrosis. Its unique advantage is that it spares the extracellular matrix, blood vessels, and nerves, thereby preserving the native tissue scaffolding and facilitating the subsequent infiltration of immune cells into the ablation zone [98]. IRE is a particularly potent activator of the cGAS-STING axis, making it highly effective at inducing strong IFN-I–driven immune priming [99]. However, the potency of this response can be limited in tumors with intrinsic suppression of the cGAS-STING pathway, a phenomenon recently documented in specific cancer subtypes [100]. The precision of the electric field is critical; incomplete electroporation can result in zones of non-immunogenic apoptosis, undermining the vaccinal effect [101,102].**High-Intensity Focused Ultrasound (HIFU)**: HIFU induces mechanical and thermal stress through acoustic cavitation and heating, generating DAMPs and exposing new epitopes. Its completely non-invasive nature is a major advantage [103]. However, its reproducibility and efficacy are highly dependent on technical factors such as acoustic windows, lesion depth, and local hemodynamics. While still largely experimental, early clinical trials in patients with hepatic metastases suggest that HIFU can synergize with ICIs to enhance T-cell infiltration and systemic cytokine responses [104].

A comparative summary of modality-specific immunologic features is provided in Table 2.

### 3.2. Antigen Dose and the Imperative of Completeness

In classical vaccines, antigen dose and formulation are critical determinants of immunogenicity. Suboptimal doses may fail to prime the immune system effectively, while excessive doses can paradoxically lead to T-cell exhaustion and tolerance. Analogous principles apply directly to ablation-based in situ vaccination.

**Incomplete Ablation**: The presence of residual viable tumor tissue is perhaps the single greatest impediment to successful in situ vaccination [105]. This residual tissue often resides in a hypoxic, nutrient-deprived state that sustains a profoundly immunosuppressive niche, dominated by regulatory T cells (Tregs), myeloid-derived suppressor cells (MDSCs), and the secretion of inhibitory cytokines like VEGF, TGF-β, and IL-10 [106]. This microenvironment can actively trap infiltrating effector T cells, induce their functional exhaustion, and prevent the establishment of systemic immunity [107]. In clinical practice, incomplete ablation has been associated with accelerated local progression and increased metastatic spread in diseases like hepatocellular carcinoma, underscoring the danger of creating an immunosuppressive “escape niche” [108].**Complete Ablation**: Ensuring the complete destruction of the target lesion is paramount. It maximizes the antigenic payload delivered to the immune system while simultaneously eliminating a major source of immunosuppressive cells and factors. Preclinical studies consistently demonstrate that complete ablation results in significantly stronger and more durable systemic T-cell responses compared to partial ablation [108]. In HCC, the completeness of ablation is a well-established key predictor of recurrence-free survival, and this benefit appears magnified when ablation is combined with adjuvant ICI therapy [107].**Antigen Integrity**: It is not only the quantity of antigen that matters, but also its quality. As mentioned, thermal denaturation induced by RFA/MWA can reduce the recognition of protein epitopes by TCRs and BCRs [108]. In contrast, modalities such as cryoablation and IRE better preserve the native conformation of antigens, potentially enhancing their immunogenicity. This is analogous to the difference between a whole-cell inactivated vaccine and a denatured protein subunit [109].

Therefore, procedural optimization—utilizing advanced intraprocedural imaging (e.g., contrast-enhanced ultrasound, CT perfusion), meticulous energy delivery mapping, and real-time thermal monitoring—is essential not only for achieving local oncologic control but also for maximizing immunologic efficacy. In this refined view, the interventional oncologist acts as an immune engineer, carefully sculpting the immune response through precise tissue destruction.

### 3.3. Route and Delivery: The Biophysical Encoding of an Immune Response

In vaccinology, the route of administration (intramuscular, subcutaneous, intradermal, mucosal) profoundly shapes the quality of the immune response by influencing antigen persistence, drainage to lymphoid organs, and the involvement of local tissue-resident immune cells [110]. For ablation, the equivalent of “route” is the biophysical nature and spatial geometry of the injury it creates.

The concentric thermal gradients characteristic of RFA/MWA generate distinct zones: a central necrotic core surrounded by a transition zone of stressed but viable cells [111]. This peripheral rim often secretes pro-angiogenic and immunosuppressive cytokines, analogous to a poorly adjuvanted vaccine that risks inducing tolerance rather than productive immunity [108].The freeze–thaw cycles of cryoablation promote necrotic death with excellent antigen preservation, akin to a whole-cell inactivated vaccine [112].The electric-field permeabilization of IRE produces a more homogeneous zone of apoptosis and necrosis while preserving the stromal architecture, analogous to a well-formulated subunit vaccine co-delivered with a potent STING-activating adjuvant [113].The acoustic cavitation and mechanical effects of HIFU can increase membrane permeability and expose new epitopes, resembling a vaccine delivered via nanoparticle or liposome technology [114].

Furthermore, the anatomical site of the tumor (e.g., hepatic vs. pulmonary vs. cutaneous) intrinsically influences the immune outcome due to differences in lymphatic drainage, baseline immune cell populations, and the organ-specific metabolic and immunologic environment [115]. For instance, liver ablation occurs in an organ rich in inherent tolerogenic signals (e.g., IL-10, TGF-β from Kupffer cells), which may dampen systemic priming unless counterbalanced by powerful adjuvant signals from the ablation itself or from combination therapy [116]. In contrast, ablation in the lung or skin—organs with robust lymphoid tissues and a predisposition towards immune surveillance—may be more conducive to the generation of potent effector and tissue-resident memory T-cell responses, akin to the advantages of mucosal vaccination routes [117].

### 3.4. Prime–Boost Scheduling: Countering Adaptive Resistance

Ablation functions most effectively as a potent priming event, generating an initial wave of antigen release and triggering the expansion of tumor-specific T-cell clones. However, tumors and their microenvironment rapidly adapt to this immune attack through a multitude of resistance mechanisms, collectively known as adaptive resistance [118,119].

These include the following:Rapid upregulation of PD-L1 on tumor cells and infiltrating myeloid cells.Recruitment and expansion of regulatory T cells (Tregs) and myeloid-derived suppressor cells (MDSCs).Induction of T-cell exhaustion phenotypes characterized by upregulation of multiple inhibitory receptors (e.g., PD-1, TIM-3, LAG-3).Secretion of inhibitory cytokines (TGF-β, IL-10).

To sustain and amplify the initial immune response, pharmacologic boosting is essential. The timing of this boost, as in classical vaccinology, is critical.

**Neoadjuvant Approach**: Administering ICIs before ablation is a strategy to “precondition” or “pre-inflame” the TME. By blocking inhibitory pathways upfront, neoadjuvant ICIs can reduce baseline immunosuppressive tone, ensuring that ablation-induced antigen release occurs in a more permissive, receptive immune environment. This strategy has shown remarkable success in the surgical neoadjuvant setting for melanoma and NSCLC [120].**Concurrent Approach**: Delivering ICIs simultaneously with ablation aims to capture the peak of innate immune activation and antigen release [121]. The goal is to prevent the immediate exhaustion of newly primed T cells by blocking PD-1/PD-L1 interactions at the moment they are first engaged, potentially leading to synergistic enhancement of CD8^+^ T cell expansion and function.**Adjuvant Approach**: Administering ICIs after ablation is designed to consolidate the immune response and prevent relapse. This approach protects the primed T-cell population from becoming exhausted as they encounter residual micrometastatic disease [122]. The phase III IMbrave050 trial in HCC, which demonstrated a significant improvement in recurrence-free survival with adjuvant atezolizumab/bevacizumab following ablation or resection, provides the strongest clinical validation for this sequencing strategy [123].

The question of optimal timing directly parallels classical prime–boost vaccination schedules, in which the interval and order of vaccinations strongly influence the balance between generating effector cells, promoting memory formation, and preventing exhaustion. Rational design of ablation–ICI regimens will require prospective, randomized trials specifically designed to test these sequencing windows [124,125].

### 3.5. Host and Tumor Microenvironmental Factors: The Biological Soil

Beyond procedure-specific variables, the host’s biological context plays a decisive role in determining the outcome of in situ vaccination.

**Baseline Immune Contexture**: The pre-existing state of the TME, as classified by the “immunoscore,” is highly predictive. Tumors with a pre-existing immune infiltrate (“hot” or “immune-inflamed” tumors) are significantly more likely to mount a robust systemic response post-ablation. In contrast, immune-desert (no infiltrate) or immune-excluded (infiltrate restricted to the stroma) tumors may require additional interventions (e.g., innate agonists, cytokines) to make the TME permissive before ablation can be effective [126].**Organ-Specific Immunology**: The liver, bone marrow, brain, and placenta are considered immunologically privileged sites, enriched in tolerogenic signals. Ablation in these locations may inherently require stronger adjuvant co-treatments to overcome the organ-imposed immune privilege [127].**Systemic Host Factors**: Comorbid conditions, concomitant medications (especially systemic corticosteroids and other immunosuppressants), age-related immunosenescence, and nutritional status can all blunt the immune response. Conversely, a diverse microbiome has been associated with improved responses to ICIs and may similarly modulate the adjuvanticity of ablation [128].**Tumor Burden and Stage**: Patients with low-volume, oligometastatic disease likely represent the ideal candidates for ablation-based vaccination [129]. The antigenic load from ablating a few lesions is sufficient to prime a response, yet the overall systemic immunosuppressive burden is not so entrenched as to be irreversible. In patients with high-volume disease, the sheer quantity of immunosuppressive factors may overwhelm any nascent immune response generated by ablating a single site [130].

## 4. Translational Strategies: Engineering a Reproducible In Situ Vaccine

The translation of tumor ablation from a cytoreductive procedure into a reliable systemic immunotherapy platform requires moving beyond opportunistic observations. In vaccinology, reproducibility is achieved through rigorous antigen standardization, controlled adjuvant delivery, and standardized prime–boost schedules. By applying these same principles, ablation can be developed into a precision in situ vaccination platform.

### 4.1. Pursuing Immune-Complete Ablation

The first and most critical determinant of immunologic efficacy is procedural completeness. Even minute volumes of residual tumor can act as immunosuppressive hubs, secreting VEGF, IL-10, and TGF-β, while attracting Tregs and MDSCs [131].

Technological innovations are increasingly focused on ensuring “immune-complete ablation,” moving beyond visual inspection to software-based and fusion imaging techniques that provide objective, reproducible confirmation of adequate margins, which is critical for robust trial outcomes [123,132,133].

**Intraprocedural imaging and margin assessment**: Advanced techniques like contrast-enhanced ultrasound (CEUS), perfusion CT, and MR thermometry can detect sublethal zones. Furthermore, the integration of CT-CT or US-MR fusion imaging and software-based volumetric analysis of the ablation zone relative to the tumor provides a more objective assessment of the minimal ablative margin (MAM) than visual inspection alone, and it directly correlates with improved local control and potentially superior immunologic outcomes [123,132,133].**Thermal Dose Mapping**: Sophisticated software algorithms now calculate the precise thermal profile of a lesion based on energy input and local tissue properties, enabling optimized and personalized energy delivery to eliminate marginal tumor cells [134].**AI-Assisted Navigation**: Machine learning algorithms are being integrated into image-guided ablation systems to improve targeting accuracy, predict heat-sink effects, and plan optimal probe trajectories, particularly in complex anatomical locations near major vessels or ducts [135].

These tools, while developed for oncologic efficacy, should also be regarded as essential immune-quality controls. Optimizing ablation is not just about preventing local recurrence but about maximizing the vaccine-like potential of the procedure.

### 4.2. Augmenting Endogenous Adjuvanticity

While DAMPs provide a natural adjuvant effect, their release is inconsistent and can be counterbalanced by the simultaneous release of tolerogenic cytokines. Rational augmentation strategies aim to boost immunogenicity and actively suppress inhibitory pathways [136].

**STING Agonists**: Direct injection of synthetic cyclic dinucleotides (e.g., ADU-S100, MK-1454) into the ablation cavity (the central zone of complete coagulative necrosis created at the treatment site following ablation) or peri-ablation zone can profoundly prolong and amplify local IFN-I production, enhancing DC licensing and T-cell priming. This is particularly relevant for tumors with intrinsic cGAS-STING suppression [137,138]. Numerous early-phase trials are testing this combination in pancreatic cancer, melanoma, and lymphoma.**TLR Agonists**: Local administration of TLR agonists, such as CpG oligodeoxynucleotides (TLR9) or poly(I:C) (TLR3), has been shown in murine models to synergize powerfully with cryoablation, leading to enhanced CD8^+^ T-cell expansion and abscopal effects [136].**Oncolytic Viruses**: Local instillation of oncolytic viruses (e.g., talimogene laherparepvec, T-VEC) after ablation can serve a dual purpose: they selectively infect and kill remaining tumor cells, and their viral pathogen-associated molecular patterns (PAMPs) provide strong adjuvant signals. Viruses engineered to express GM-CSF can further enhance local DC recruitment and activation [137].**Nanoparticle-Based Adjuvants**: The development of injectable, thermoresponsive biomaterials and liposomes represents a cutting-edge approach. These can be designed to release immunostimulatory payloads (e.g., cytokines, agonists) in a sustained manner within the ablation cavity, effectively creating a long-lasting “immune niche” that prolongs immune activation beyond the acute phase of cell death [138].

These strategies reflect the core vaccinology principle that an antigen without adequate adjuvant signaling is rarely sufficient for productive immunity. By combining ablation with rational adjuvant engineering, the goal of reproducible systemic immunity becomes increasingly feasible.

### 4.3. A Biomarker Framework for Vaccination Efficacy

Biomarkers are the essential bridge between mechanism and clinical translation. The objective is to develop immune equivalents of vaccine titers, enabling standardized assessment, patient stratification, and regulatory approval.

A consolidated overview of candidate biomarkers and their rationale for ablation-induced in situ vaccination is presented in Table 3.

#### 4.3.1. Pharmacodynamic Biomarkers (Measure the “Activity” of the Vaccine)

Serum HMGB1, ATP (24–72 h post-ablation): Quantifies the acute release of key DAMPs and serves as a direct measure of ICD magnitude [139].IFN-I Signatures (ISG15, IFIT1, MX1, IFI44; Day 3–10): A peripheral blood mononuclear cell (PBMC) transcriptional score or serum protein level of IFN-α/β can serve as a robust readout of systemic innate immune activation, a critical step for T-cell priming [140].Serum IL-6, VEGF: These can serve as negative pharmacodynamic biomarkers, indicating the presence of sublethal injury and a shift towards an immunosuppressive, pro-angiogenic bias [141].

#### 4.3.2. Immunological Biomarkers (Measure the “Response” to the Vaccine)

TLS Detection and Scoring (Weeks 4–6 post-ablation):Multiplex immunohistochemistry (IHC) and spatial transcriptomics (e.g., GeoMx, CosMx) should be performed approximately 4–6 weeks after ablation (relative to Day 0) to quantify TLS architecture, cellular composition, and maturation state (e.g., presence of germinal centers). Mature TLSs observed at this post-ablation stage are a strong positive predictor of durable immune response [142].TCR Repertoire Analysis (Weeks 2–4 post-ablation):High-throughput TCR sequencing (TCR-Seq) of peripheral blood or tumor tissue obtained 2–4 weeks after ablation (relative to Day 0) allows tracking of clonal expansion and the breadth of epitope spreading. An increase in T-cell clonality and richness during this early adaptive-response window post-ablation is a favorable immunologic sign [143].Functional T-cell Assays (Weeks 4–6 post-ablation):Ex vivo stimulation of PBMCs with tumor lysate or peptide pools performed approximately 4–6 weeks after ablation (relative to Day 0) can quantify the frequency and polyfunctionality (production of IFN-γ, TNF-α, IL-2) of tumor-reactive T cells [144].

#### 4.3.3. Tumor-Derived Biomarkers

ctDNA Clearance (2–6 weeks post-ablation, relative to Day 0): The reduction and eventual clearance of circulating tumor DNA (ctDNA) within 2–6 weeks after ablation (Day 0) represents a highly promising, non-invasive biomarker for predicting systemic tumor control and recurrence-free survival in colorectal cancer, hepatocellular carcinoma (HCC), and other malignancies [145]. This interval reflects the expected window of early immune-mediated clearance of minimal residual disease.Tumor-Derived Exosomes (1–4 weeks post-ablation, relative to Day 0): Shifts in the protein or RNA cargo of tumor-derived exosomes isolated from plasma approximately 1–4 weeks after ablation (Day 0) can provide an early readout of tumor immunogenicity changes and stress-response signaling [146]. Such changes frequently precede measurable clinical or radiologic responses.Tumor-Intrinsic Biomarkers (Baseline/pre-ablation; Day −7 to 0): Assessment of tumor expression of innate-immune suppressors, such as RECQL4 or TRIM6, should be performed before ablation (Day −7 to Day 0) to identify patients who may be intrinsically resistant to ablation-induced STING activation and who could benefit from combination therapy with specific innate agonists [147].

Collectively, these biomarkers form the backbone of a vaccinology-inspired monitoring framework that can guide therapy and assess efficacy.

### 4.4. Standardized Assay Timepoints for Cross-Trial Interpretation

Without harmonized timing, biomarker data remain incomparable across trials and institutions. To ensure consistency, all hours, days, and months in this framework are referenced to the day of ablation (Day 0), which serves as the procedural baseline for immunologic and biomarker assessments. We propose the following standardized blood and tissue sampling schedule:Baseline (Pre-ablation; Day −7 to 0): Comprehensive immune profiling (flow cytometry), circulating tumor DNA (ctDNA) measurement, serum cytokines, and tumor biopsy for immune contexture and innate immune suppressor status (e.g., RECQL4, TRIM6).24–72 h post-ablation (relative to Day 0): Peak release of damage-associated molecular patterns (DAMPs), including serum HMGB1 and ATP, accompanied by early cytokine changes (IL-6, VEGF, IL-10).Day 7–10 post-ablation: Period of peak innate immune activation characterized by elevated type I interferon (IFN-I) transcriptional signatures in peripheral blood mononuclear cells (PBMCs) and serum CXCL10.Week 2–4 post-ablation: Early adaptive immune-response phase, reflected by expansion of tumor-specific T-cell clones (TCR clonality in blood) and initial ctDNA kinetics.Week 4–6 post-ablation: Established adaptive immune response and early memory formation, assessed by comprehensive immune profiling, functional T-cell assays, and ctDNA clearance.Follow-up at 3, 6, and 12 months post-ablation: Long-term immune-memory and surveillance phase, including serial ctDNA monitoring and delayed assessment of tertiary lymphoid structure (TLS) formation on imaging if recurrence occurs.

This structured design mirrors vaccine immunogenicity studies, which measure antibody titers and T-cell assays at specific intervals post-vaccination [148]. Standardized immunologic windows will dramatically accelerate cross-trial interpretability and facilitate meta-analyses.

### 4.5. Clinical Trial Design: From Exploration to Confirmation

To date, most studies combining ablation with immunotherapy are exploratory, single-arm, or retrospective. A definitive roadmap for robust clinical translation must include:

#### 4.5.1. Endpoint Reformation

Moving beyond local control, trials should prioritize non-index lesion endpoints, using immune-modified criteria such as irRECIST to capture abscopal effects. Recurrence-free survival (RFS) and ctDNA clearance are highly relevant surrogate endpoints. Local control should be rigorously assessed using standardized volumetric software to confirm ablation completeness, a factor now shown in randomized trials to be critical for outcomes [149,150,151].

#### 4.5.2. Mandatory Biomarker Integration

Pre-specified biomarker panels should be a mandatory component of trial design, not an exploratory add-on. This includes not only serum and immune biomarkers but also tumor-intrinsic biomarkers of immune competence, such as the expression of proteins that regulate key pathways, like cGAS-STING (e.g., RECQL4, TRIM6), which may stratify patients who are likely to need additional innate immune agonists [152,153].

#### 4.5.3. ICI Timing Randomization

Phase II trials should randomize patients to neoadjuvant, concurrent, or adjuvant ICI arms to definitively resolve the question of optimal sequencing.

#### 4.5.4. Window-of-Opportunity Cohorts

Embedding small, intensive biomarker-monitoring cohorts within larger trials enables rapid qualification of predictive and pharmacodynamic biomarkers, including novel imaging biomarkers such as LAG-3 PET to track immune cell infiltration [154].

#### 4.5.5. Multimodal Combinations

Exploring triple combinations—e.g., ablation + radiotherapy (for enhanced antigen release) + ICI, or ablation + oncolytic virus + TLR agonist—will be necessary to overcome resistance in “cold” tumors, particularly those with biomarkers indicating suppressed innate immunity [155,156].

Such rigorously designed trials are essential to determine whether ablation can consistently be transformed from a local tool into a vaccine-like systemic therapeutic modality.

## 5. Clinical Applications and Trials

Building on the immunologic principles outlined in the preceding sections, this chapter discusses how tumor ablation can be integrated with systemic immunotherapies to optimize both local control and systemic antitumor immunity. The goal is to translate the in situ vaccination effect into consistent clinical benefit by combining ablation with immune-checkpoint inhibitors, cytokine modulators, and pattern-recognition receptor agonists. Various temporal and mechanistic strategies—such as neoadjuvant, concurrent, and adjuvant approaches—are explored, together with emerging clinical trial frameworks that aim to standardize these interventions. Collectively, these approaches represent the next stage in the evolution of ablation from a local procedure to a reproducible immunotherapeutic platform.

### 5.1. Clinical Translation Across Tumor Types

The clinical trial landscape is rapidly expanding (Figure 1).

Table 4 summarizes selected key trials investigating this paradigm. Notable trends include the dominance of HCC trials (reflecting its global burden and the synergy between local and systemic therapy), the prominence of cryoablation in breast and prostate cancer, and the exploration of novel adjuvants (TLR agonists, STING agonists) in early-phase studies.

### 5.2. Safety and Nuanced Patient Selection

While generally well-tolerated, ablation carries modality-specific risks that intersect with immunologic goals [157,158].

#### 5.2.1. Cryoablation

The robust inflammatory response is immunologically beneficial but carries a rare risk of cryoshock syndrome, a systemic inflammatory response with coagulopathy and multi-organ failure. Meticulous technique and complete freeze–thaw cycles mitigate this risk [159,160].

#### 5.2.2. RFA/MWA

Risks include pain, post-ablation syndrome (fever, malaise), and incomplete ablation due to the heat-sink effect. The sublethal rim can be a source of immunosuppression [161,162].

#### 5.2.3. IRE

Requires general anesthesia and precise ECG synchronization to avoid potentially fatal arrhythmias. Its non-thermal, matrix-sparing nature is a significant immunologic advantage [163,164].

#### 5.2.4. HIFU

Non-invasive, but can be limited by bone or air interposition [165]. Risks include skin burns and cavitation-induced damage to adjacent structures [166].

Patient selection must consider both oncologic and immunologic factors. Key contraindications include active uncontrolled infections, ongoing flares of severe autoimmune disease requiring immunosuppression, and concurrent use of high-dose systemic corticosteroids. Ideal candidates are likely those with oligometastatic or low-volume metastatic disease, good performance status, and evidence of a competent immune system [167].

## 6. Limitations and Outlook

Despite the compelling rationale and encouraging early data, several significant limitations currently hinder the reproducibility and broad adoption of ablation as a reliable in situ vaccination strategy.

### 6.1. Biological and Mechanistic Heterogeneity

The immune consequences of ablation are not uniform. Identical procedures can yield opposite outcomes depending on factors such as tumor histology, genetic makeup, and baseline immune contexture [168]. For instance, while cryoablation preserves antigenicity, it can also release tolerogenic extracellular vesicles in some contexts [169]. Similarly, the ability to form functional TLS post-ablation is highly variable. Furthermore, tumor-intrinsic suppressors of immune sensor pathways have emerged as key predictive biomarkers. For example, high expression of RECQL4 in HCC or TRIM6 in microsatellite-stable gastric cancer potently suppresses the cGAS-STING pathway, leading to poor T-cell infiltration and likely blunting the response to ablation-driven vaccination without additional targeted modulation [138].

### 6.2. Clinical and Technical Variability

A major obstacle is the lack of procedural standardization. Energy delivery parameters, device manufacturers, operator skill, and lesion characteristics (size, location) vary immensely across studies. This heterogeneity makes cross-trial comparisons and meta-analyses exceedingly difficult, slowing down the generation of high-level evidence needed for guidelines [83].

### 6.3. Biomarker Validation and Access

While promising biomarkers exist, none have been prospectively validated or approved for clinical use. Liquid biopsies (ctDNA) are scalable but require standardization. Advanced spatial profiling techniques (multiplex IHC, spatial transcriptomics) provide deep mechanistic insights but are costly, complex, and not widely available, limiting their use in multicenter trials [170].

### 6.4. Trial Design Gaps

Many existing studies are limited by their single-arm, exploratory nature. Common shortcomings include the following:Lack of randomization for ICI timing.Absence of control arms (ablation alone vs. ablation + ICI).Underpowered sample sizes for assessing systemic efficacy.Inconsistent use of immune-related response criteria.Inadequate integration of correlative biomarker studies.

## 7. Conclusions

Tumor ablation is evolving from a local cytoreductive technique into a reproducible in situ vaccination platform; by ensuring procedural completeness, integrating immune checkpoint blockade or innate agonists—particularly in settings with cGAS–STING pathway suppression such as RECQL4-high HCC or TRIM6-high MSS gastric cancer—and embedding a robust biomarker framework, the abscopal effect can be transformed from a rare anecdote into predictable systemic immunity, positioning ablation as a cornerstone of next-generation cancer immunotherapy.

A graphical summary of the mechanistic and translational concepts discussed in this review is presented in Figure 2, highlighting the sequential immunological events and key therapeutic intervention points.

## Figures and Tables

**Figure 1 vaccines-13-01114-f001:**
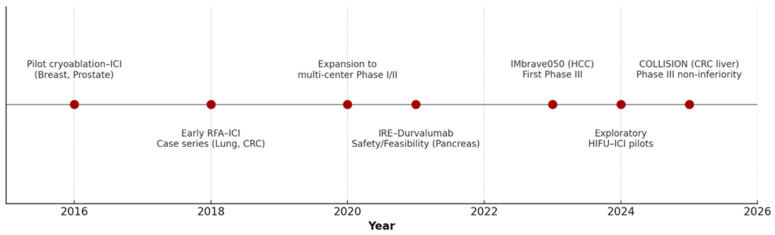
Timeline of Clinical Development of In Situ Vaccination by Tumor Ablation from 2015 to 2025.

**Figure 2 vaccines-13-01114-f002:**
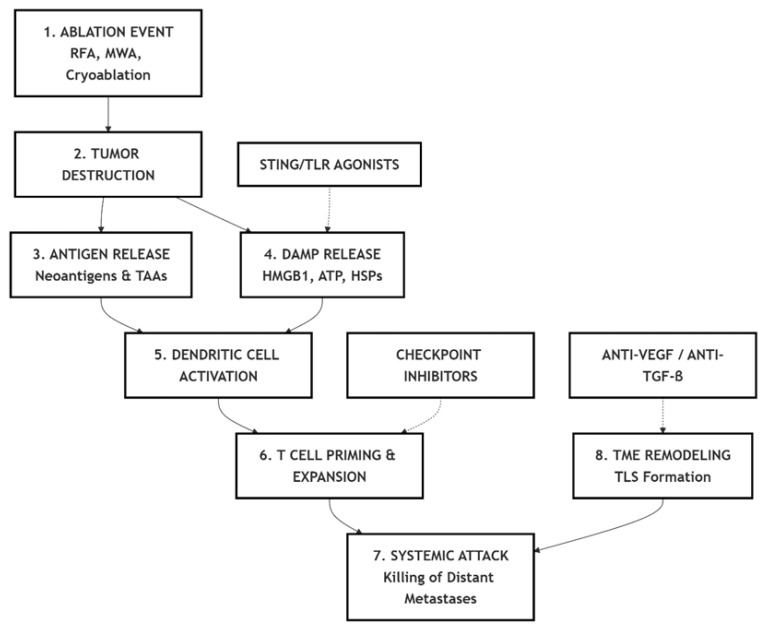
Summary schematic of the immunological cascade and therapeutic modulation in in situ vaccination induced by tumor ablation. RFA—Radiofrequency ablation; MWA—Microwave ablation; TAA—Tumor-associated antigen; DAMP—Damage-associated molecular pattern; HMGB1—High-mobility group box 1; ATP—Adenosine triphosphate; HSP—Heat shock protein; TME—Tumor microenvironment; TLS—Tertiary lymphoid structure; STING—Stimulator of interferon genes; TLR—Toll-like receptor; VEGF—Vascular endothelial growth factor; TGF-β—Transforming growth factor beta.

**Table 1 vaccines-13-01114-t001:** Comparison Between Classic Systemic Vaccination and In Situ Vaccination by Tumor Ablation.

Parameter	Classic Systemic Vaccination	In Situ Vaccination by Tumor Ablation
**Antigen Source**	Exogenous, predefined antigens (e.g., peptides, recombinant proteins, DNA/RNA vaccines) designed or selected based on known tumor-associated or neoantigen sequences.	Endogenous, patient-specific antigens derived from the patient’s own tumor cells released during ablation-induced immunogenic cell death (ICD).
**Adjuvant**	Externally added immune stimulants (e.g., CpG, GM-CSF, poly I:C, alum) to promote antigen-presenting cell (APC) activation.	Endogenously generated damage-associated molecular patterns (DAMPs) such as calreticulin, ATP, HMGB1, and DNA that activate innate immunity through PRRs and the cGAS–STING pathway.
**Antigen Presentation Site**	Occurs in secondary lymphoid organs following systemic administration of antigen/adjuvant formulation.	Initiated locally within the ablation site and its draining lymph nodes, often accompanied by tertiary lymphoid structure (TLS) formation.
**Delivery Route**	Intramuscular, subcutaneous, intradermal, or intravenous injection.	Local tumor ablation via thermal (RFA, MWA, HIFU), cryogenic, or nonthermal (IRE, laser, photodynamic) modalities.
**Mechanism of Action**	Prepares the immune system against anticipated antigens through controlled exposure and immune priming.	Converts the tumor itself into a personalized “vaccine depot,” combining antigen release and in situ adjuvant generation in a natural microenvironment.
**Immune Activation**	Primarily adaptive (T- and B-cell) immunity; innate activation depends on adjuvant potency.	Potent innate and adaptive immune activation through ICD, DAMP signaling, and antigen cross-presentation by dendritic cells.
**Personalization**	Requires prior knowledge of tumor antigens or sequencing data; standardized formulation across patients.	Fully personalized; antigens reflect each patient’s unique mutational and post-translational landscape.
**Advantages**	High controllability of antigen/adjuvant dose and schedule.—Safe, standardized, and scalable production.	No need for prior antigen identification.—Broad antigenic repertoire and potential to induce epitope spreading.—Can synergize with immune checkpoint inhibitors (ICIs).
**Disadvantages/Limitations**	Limited antigen breadth; risk of immune escape.—May fail to generate strong cytotoxic T-cell responses in immunosuppressive TME.	Heterogeneous immune activation depends on ablation completeness, modality, and tumor contexture.—Procedural variability and local immunosuppression may limit systemic efficacy.
**Clinical Endpoint**	Serological or T-cell immune response and protection against target antigen or pathogen/tumor.	Local tumor control, induction of systemic (abscopal) responses, and conversion of “cold” tumors to “hot.”

**Table 2 vaccines-13-01114-t002:** Immunologic profiles of ablation modalities.

Modality	Mechanism	Antigen Integrity	DAMP Profile	Immune Risk	Immune Signature (Typical)	Clinical Notes
RFA/MWA	Thermal coagulation	Denatured proteins; broad antigen release	ATP, HMGB1, ROS	Sublethal rim → IL-6/VEGF/HGF	High DAMPs; rim-related tolerance risk	Common in liver/kidney
Cryoablation	Freeze–thaw necrosis	Preserved epitopes	Strong inflammatory response	Risk of incomplete freeze	Preserved epitopes; type-1-skewed inflammation	Breast, lung, bone
IRE	Electric membrane disruption	Largely intact antigen repertoire	ICD + cGAS–STING activation	Requires precise field coverage	STING-skewed ICD; vascular/matrix sparing	Pancreas, liver
HIFU	Focused ultrasound	Variable (site/depth dependent)	DAMPs + cavitation effects	Depth-/window-dependent	Cavitation-linked DAMPs; variable	Experimental/selected centers

*Abbreviations*: RFA, radiofrequency ablation; MWA, microwave ablation; IRE, irreversible electroporation; HIFU, high-intensity focused ultrasound; DAMP, damage-associated molecular pattern; ROS, reactive oxygen species; ICD, immunogenic cell death.

**Table 3 vaccines-13-01114-t003:** Candidate biomarkers for ablation-induced in situ vaccination.

Category	Biomarker	Rationale/Clinical Notes
**Pharmacodynamic (early innate activation)**	Serum HMGB1, ATP (24–72 h after ablation)	Direct readouts of immunogenic cell death (ICD) magnitude correlate with dendritic cell (DC) licensing.
	Whole-blood IFN-I transcriptional signatures (ISG15, IFIT1, MX1, IFI44) (day 3–10 post-ablation)	Surrogate for STING pathway activation and systemic innate priming.
	Serum IL-6, VEGF, IL-10 (24–72 h after ablation)	Negative biomarkers: indicate sublethal injury, pro-angiogenesis, and immunosuppressive skew.
**Immunological (adaptive response)**	TLS formation and maturation (multiplex IHC, spatial transcriptomics) (weeks 4–6 post-ablation)	Presence and organization of TLS predict sustained immune priming, improved ICI response, and survival.
	TCR repertoire clonality/expansion (TCR-seq) (week 2–4 post-ablation)	Tracks epitope spreading and diversity of anti-tumor T-cell response post-ablation.
	Functional T-cell assays (ex vivo cytokine production: IFN-γ, TNF-α, IL-2) (week 4–6 post-ablation)	Measure the breadth and polyfunctionality of tumor-reactive T cells in the blood or the tumor.
	B-cell responses/antibody repertoire shifts (month 1–3 post-ablation)	Report on humoral contribution to in situ vaccination, often linked to TLS activity.
**Tumor-Derived/Liquid Biopsy**	ctDNA kinetics and clearance (2–6 weeks after ablation)	Sensitive, non-invasive marker of minimal residual disease and systemic tumor control.
	Tumor-derived exosome profiling (RNA/protein cargo) (weeks 1–4 post-ablation)	Reflects tumor stress and changing immunogenicity post-ablation.
	Circulating tumor cells (CTCs) (weeks 2–6 post-ablation)	Dynamic measure of systemic disease burden; reduction correlates with immune-mediated clearance.
**Imaging Biomarkers**	LAG-3 or PD-L1 PET imaging (weeks 4–6 after ablation)	Enables real-time tracking of immune infiltration and checkpoint dynamics post-ablation.
	Radiomics (AI-guided analysis of pre-/post-ablation imaging)	Predicts immune responsiveness, completeness of ablation, and likelihood of abscopal effect.
**Tumor-Intrinsic/Predictive**	RECQL4, TRIM6 expression (baseline, pre-ablation)	Predict resistance to cGAS-STING activation; identify patients needing innate agonist combinations.
	Tumor mutational burden (TMB), neoantigen load (baseline, pre-ablation)	Predicts likelihood of durable systemic immunity; higher burden favors ablation + ICI synergy.
	Baseline immune contexture (inflamed vs. excluded vs. desert phenotype) (pre-ablation biopsy)	Determines the TME’s readiness to respond; immune-desert tumors may require adjuvant modulation.

**Table 4 vaccines-13-01114-t004:** Clinical trials investigating ablation as in situ vaccination (2022–2025).

Tumor Type	Ablation Modality	Combination Therapy	Trial Phase/ID	Region/Status	Key Findings/Notes	ICI Timing
Hepatocellular carcinoma (HCC)	RFA/MWA/resection	Atezolizumab + Bevacizumab	Phase III, IMbrave050 (NCT04102098)	Global/Completed	Improved recurrence-free survival; established proof of adjuvant ICI post-ablation	Adjuvant
Hepatocellular carcinoma	MWA	Camrelizumab (PD-1)	Phase II, NCT04566133	China/Ongoing	Evaluating recurrence-free survival and immune activation	NR
Hepatocellular carcinoma	RFA	Tremelimumab (CTLA-4) + Durvalumab	Phase I/II, NCT02821754	Multicenter/Ongoing	Preliminary evidence of immune activation; feasibility	NR
Breast cancer	Cryoablation	Pembrolizumab (PD-1)	Phase II, NCT03546686	US/Active	Enhanced systemic T-cell priming; ongoing accrual	NR
Breast cancer	Cryoablation	Ipilimumab (CTLA-4)	Pilot, NCT01992250	US/Completed	TLS induction; systemic immune activation observed	NR
Breast cancer (locally advanced)	Cryoablation	TLR9 agonist + PD-1 blockade	Phase I/II, NCT04698187	US/Ongoing	Early data suggest enhanced adjuvanticity and TLR synergy	NR
Lung cancer (NSCLC)	RFA/Cryoablation	Nivolumab ± Ipilimumab	Phase II, NCT03275597	Europe/Interim	Abscopal-like responses reported in subset of patients	NR
Pancreatic cancer	IRE	Durvalumab (PD-L1)	Phase I/II, NCT03753680	US/Recruiting	Increased immune infiltration; technically feasible	NR
Pancreatic cancer	IRE	Nivolumab ± chemotherapy	Pilot, NCT04301778	China/Active	Evaluating safety and immunogenicity; early biomarker data pending	Concurrent
Colorectal metastases (liver)	RFA	Tremelimumab (CTLA-4) + Durvalumab	Phase I/II, NCT02821754	Europe/Ongoing	Early immune activation signals	NR
Prostate cancer	Cryoablation	Ipilimumab (CTLA-4)	Pilot, NCT01961636	US/Completed	Demonstrated feasibility; modest immune modulation	NR
Renal cell carcinoma	Cryoablation	Nivolumab (PD-1)	Phase II, NCT02833233	US/Recruiting	Assessing immune modulation and abscopal activity	NR
Melanoma (advanced)	Cryoablation	TLR9 agonist + PD-1 blockade	Phase I/II, NCT04698187	Multicenter/Active	Investigating local adjuvant boosting	NR
Liver metastases (CRC, breast)	HIFU	Anti–PD-1/PD-L1 agents	Pilot, China (2023–2024)	Early feasibility	Enhanced infiltration and cytokine response; safety acceptable	Concurrent
Gynecologic cancers (ovarian, cervical)	Cryoablation	Pembrolizumab (PD-1)	Phase I, NCT05270882	US/Recruiting	Testing feasibility and safety; biomarker endpoints	NR
Mixed solid tumors	Laser thermal ablation	Nivolumab + STING agonist	Early-phase exploratory study (2024)	Europe/Initiating	Novel combination; assessing DAMP release and IFN-I signatures	Concurrent

*Abbreviations*: ICI, immune checkpoint inhibitor; NR, not reported; RFA, radiofrequency ablation; MWA, microwave ablation; IRE, irreversible electroporation; HIFU, high-intensity focused ultrasound; TLS, tertiary lymphoid structure; TLR, Toll-like receptor.

## Data Availability

No new data were created or analyzed in this study.

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
