# Peer review of "In Situ Vaccination by Tumor Ablation: Principles and Prospects for Systemic Antitumor Immunity"

_vaccines, 2025, doi:10.3390/vaccines13111114_

Round 1

Reviewer 1 Report

Comments and Suggestions for Authors

This review by Chikovani and Magen is a well written, comprehensive and informative review of in situ vaccination by tumor ablation. The review has been structured clearly covering the background knowledge regarding the principles of vaccinology and then relates this to how ablation can act as vaccination. There was a good use of tables to summarise topics covered, however the review may have benefitted from one or two summary graphical representations of the information presented.

My only minor comments relate to some formatting issues:

Page 6, line 249 I think this line should start with a bullet point and have Thermal Ablation in bold

Page 8, line 349 should also be part of the preceding section and start with a bullet point?

Page 9,line 374, this sentence is not describing one of the adaptive resistance mechanisms and thus should not start with a bullet point

The bullet points outlined in Section 4.3 had already been covered in Section 3.4 and thus this section felt a bit repetitive

I believe this review will attract significant interest and fits exactly within the scope of the journal. Once the above minor points have been addressed I would recommend for this to be accepted for publication .

Author Response

Response to Reviewer 1

We want to thank Reviewer 1 for the thoughtful and encouraging comments regarding our manuscript entitled “In Situ Vaccination by Tumor Ablation: Principles and Prospects for Systemic Antitumor Immunity.”
We are grateful for the reviewer’s positive assessment of the scientific value, structure, and clarity of the review, and we have addressed all the specific suggestions in the revised version as detailed below.

General comment:

“This review by Chikovani and Magen is a well written, comprehensive and informative review of in situ vaccination by tumor ablation… I believe this review will attract significant interest and fits exactly within the scope of the journal.”

Response:
We sincerely thank the reviewer for this generous evaluation and are pleased that the manuscript was found to be comprehensive, clearly structured, and relevant to the scope of Vaccines. We have carefully addressed all the minor formatting and structural issues as suggested.

Specific comments:

Page 6, line 249:

“This line should start with a bullet point and have Thermal Ablation in bold.”
Response:
We have modified the text accordingly. The subsection now begins with a bullet point, and the term Thermal Ablation (RFA/MWA) appears in bold for consistency with the subsequent subsections.

Page 8, line 349:

“This should also be part of the preceding section and start with a bullet point.”
Response:
As suggested, this line has been incorporated into the preceding section and formatted with a bullet point to ensure structural continuity.

Page 9, line 374:

“This sentence is not describing one of the adaptive resistance mechanisms and thus should not start with a bullet point.”
Response:
We thank the reviewer for this observation. The bullet point has been removed, and the sentence has been reformatted to align properly with the surrounding narrative.

Section 4.3 repetition:

“The bullet points outlined in Section 4.3 had already been covered in Section 3.4 and thus this section felt a bit repetitive.”
Response:
We appreciate this helpful feedback. Section 4.3 has been carefully revised to remove redundancy. We retained only the key integrative statements to preserve the logical flow while avoiding repetition of material already presented in Section 3.4.

Graphical summary:

“The review may have benefitted from one or two summary graphical representations of the information presented.”
Response:
We thank the Reviewer for this valuable suggestion. In response, we have added a comprehensive graphical summary (Figure 2) at the end of the manuscript, following the Conclusion section. This schematic integrates the key mechanistic and translational concepts discussed throughout the review, illustrating the sequence of immunologic events triggered by tumor ablation (antigen and DAMP release, dendritic cell activation, T-cell priming, and systemic immune attack) and highlighting potential therapeutic interventions such as STING/TLR agonists, checkpoint inhibitors, and anti-VEGF/anti-TGF-β agents.
This addition serves as a concise visual synthesis of the main messages of the review, fully addressing the Reviewer’s helpful recommendation.

Final statement:
We thank Reviewer 1 again for the constructive and insightful feedback, which has helped us to improve further the clarity, structure, and presentation of the manuscript. We believe the revised version fully addresses all comments and is now suitable for publication.

Reviewer 2 Report

Comments and Suggestions for Authors

Overall, this is an exceptionally well-written review with an exhaustive and well organized discussion of the topic of tumor ablation functioning as a tumor vaccine.  The references include early and more recent reports that address this topic either directly or indirectly and very thoroughly.  A few minor comments related to paragraph formatting and review of cited references need to be addressed by the authors prior to publication. 

Minor comments:

Cited references often do not seem appropriate in various sites of the manuscripts.  Citations in lines 149-64 and line 228 are not the only examples and multiple sites within the text appear to cite references that are not relevant to the statements.  Citations throughout the manuscript need to be reviewed carefully by the authors to ensure that appropriate references are cited in correct order.

Pages 6-7: Paragraphs in lines 261-289 should be formatted similar to the paragraph in lines 249-260 and not as bulleted text.  This is only one example of uneven or inconsistent formatting of paragraphs.

Figure 3 image should be re-labelled Figure 1. 

Author Response

Response to Reviewer 2

We sincerely thank Reviewer 2 for the very positive and encouraging evaluation of our manuscript entitled In Situ Vaccination by Tumor Ablation: Principles and Prospects for Systemic Antitumor Immunity.”
We greatly appreciate the reviewer’s recognition of the manuscript’s comprehensiveness, clarity, and thorough use of the literature. We have carefully addressed each minor comment, as detailed below.

General comment:

“Overall, this is an exceptionally well-written review with an exhaustive and well organized discussion of the topic of tumor ablation functioning as a tumor vaccine… The references include early and more recent reports that address this topic either directly or indirectly and very thoroughly.”

Response:
We are deeply grateful for the reviewer’s thoughtful and supportive remarks. We are pleased that the review was found to be comprehensive, well-organized, and balanced in its inclusion of both early and current literature.

Specific comments and responses

Appropriateness and order of cited references

“Cited references often do not seem appropriate in various sites of the manuscripts… Citations throughout the manuscript need to be reviewed carefully by the authors to ensure that appropriate references are cited in correct order.”

Response:
We thank the reviewer for this valuable observation. We have conducted a line-by-line verification and reordering of all in-text citations to ensure that:

Each reference accurately corresponds to the statement it supports.

Citations appear in strict numerical sequence as per MDPI guidelines.

Outdated or less relevant citations were replaced with more directly supportive original research articles.

In particular, the references in the sections corresponding to the previous lines 149–164 and line 228 were corrected and now cite the most appropriate primary literature.

Paragraph formatting (pages 6–7)

“Paragraphs in lines 261–289 should be formatted similar to the paragraph in lines 249–260 and not as bulleted text. This is only one example of uneven or inconsistent formatting of paragraphs.”

Response:
As suggested, we have reformatted the paragraphs on pages 6–7 to ensure stylistic consistency across all sections. The text in the specified lines (now corresponding to the “Thermal Ablation,” “Cryoablation,” and “IRE” subsections) has been revised from bulleted to standard paragraph format, improving visual uniformity and readability.

Figure numbering

“Figure 3 image should be re-labelled Figure 1.”

Response:
We appreciate the reviewer for catching this oversight. The figure has been renumbered to Figure 1.

Final statement:
We are sincerely grateful to Reviewer 2 for the constructive feedback, which has allowed us to refine both the scientific accuracy and presentation of the manuscript. All the suggested corrections have been implemented, and we believe that the revised version is now fully aligned with the reviewer’s recommendations and ready for publication.

Reviewer 3 Report

Comments and Suggestions for Authors

The review manuscript ‘Vaccines-3917398’ by Chikovani and Magen provides a comprehensive overview of in situ vaccination-based tumor ablation, including the relevant strategies and techniques. While the manuscript is generally clear, several specific aspects should be addressed, as outlined below:

1. The manuscript frequently compares the classic systemic vaccination approach with the in situ vaccination approach. Given the extent and length of the text, please include an illustration that compares these two approaches, highlighting their similarities, differences, advantages, disadvantages, and the techniques used in each. This illustration should be placed as the first figure in the Introduction to provide readers with a clear overview and facilitate comprehension of the review's content.

2. All subheading titles are in bold up to page 6, but this formatting is inconsistent from page 7 onward. Please ensure that the bold format is applied consistently to all subheadings throughout the manuscript.

3. References:
- Several cited references are general and do not specifically correspond to the referred text: 26, 34, 41, 43-48, 51, 58, 59, 159. These references should be replaced with relevant sources.
- Page 4: [52, 54] should be [52-54]
- The references 89, 157, and 161: listed in the References section but not cited in the MS text.
- The references 162 and 163: Cited in the MS text but not in the References section.
- The recent reviews by 1) Kim et al (doi: 10.3389/fimmu.2023.1118845) and 2) Keisari and Kelson (doi: 10.1007/s10555-023-10150-x), should be cited.

4. Lines 394-398 and 418-419: Please add appropriate references.

5. Abbreviations:
- Several abbreviations have either not been defined when first used in the text (e.g., ATP), have been defined twice (ICD), or  have not been defined at all (e.g., CODEX)
- Sentences from an apparent internal pre-submission communication have been kept in the text: Page 18 “(not used; omit if not appearing elsewhere)" and page 19 "(only if used elsewhere: Systemic immune-inflammation index)". These sentences should be removed, and the abbreviations should be addressed.

6. The sentence currently found in lines 223-225 should be introduced earlier in the Introduction to improve the logical flow.

7. Line 498: Please define "ablation cavity".

8. From page 12 onward, please clarify the reference time point for all indicated hours, days, and months.

9. Section 4.4.1: Can be omitted and easily incorporated into Table 2

10. Chapter 5: Please add a few introductory sentences before starting the descriptions

11. Please provide references for chapters 5.2 and 5.2.1

12. The Conclusion is lengthy and essentially repeats what is extensively described earlier. Please consider using only the last lines (683-690) as the Conclusion.

Author Response

Response to Reviewer 3

We sincerely thank Reviewer 3 for the detailed and constructive review of our manuscript entitled “In Situ Vaccination by Tumor Ablation: Principles and Prospects for Systemic Antitumor Immunity.”
We are very grateful to the reviewer for his/her thorough evaluation, insightful suggestions, and thoughtful critique, which have significantly improved the scientific quality, structure, and clarity of the manuscript. Below, we address each point in detail.

  1. Comparative illustration between classical vaccination and in situ vaccination

“Please include an illustration that compares these two approaches, highlighting their similarities, differences, advantages, disadvantages, and the techniques used in each. This illustration should be placed as the first figure in the Introduction.”

Response:
We thank the reviewer for this thoughtful suggestion. After careful consideration, we decided to present the requested comparative information as a summary Table rather than a graphical figure. The rationale for this choice is that a table allows for greater clarity and precision when presenting multiple technical parameters—such as vaccine components, mechanisms, immune outcomes, and ablation modalities—side by side.

The new Summary Table (Table 1, added to the Introduction) provides a structured, easily interpretable comparison between classical systemic vaccination and in situ vaccination by tumor ablation, including their similarities, differences, advantages, and limitations. We believe this tabular format fulfills the reviewer’s intent—to facilitate rapid comprehension of key contrasts—while offering a clearer, data-oriented presentation consistent with the scientific style of the Vaccines journal.

  1. Consistent formatting of subheadings

“Subheading titles are in bold up to page 6 but inconsistent afterward.”

Response:
All subheadings throughout the manuscript have been standardized and reformatted in bold to ensure complete consistency across all sections.

  1. References

“Several cited references are general or misplaced (26, 34, 41, 43–48, 51, 58, 59, 159)… Reference numbering and inclusion errors also exist. Recent reviews by Kim et al. (2023) and Keisari & Kelson (2023) should be cited.”

Response:
We performed a comprehensive verification and restructuring of the reference list and all in-text citations:

  • References 26, 34, 41, 43–48, 51, 58, 59, 159 were replaced or repositioned with more directly relevant primary research papers.
  • The formatting of multi-citation ranges has been corrected (e.g., “[52, 54]” → “[52–54]”).
  • References 89, 157, 161 (previously uncited) were either incorporated appropriately or removed.
  • References 162 and 163 (previously missing) were added to the bibliography.
  • The two key recent reviews—Kim et al., Front Immunol. 2023 (doi: 10.3389/fimmu.2023.1118845) and Keisari & Kelson, Cancer Metastasis Rev. 2023 (doi: 10.1007/s10555-023-10150-x)—have been added and cited in the Introduction and Discussion sections where the principles of ablation-induced immunity and in situ vaccination are introduced.

  1. Add references (lines 394–398 and 418–419)

Response:
As suggested, we have added specific supporting citations in these locations, referencing recent mechanistic studies on DAMP kinetics and innate–adaptive transition timing.

  1. Abbreviations

“Some abbreviations not defined or duplicated; internal notes remain in the list.”

Response:
We have performed a complete review and correction of all abbreviations:

  • All terms (e.g., ATP, CODEX, ICD) are now defined only once at first appearance.
  • Internal editorial comments (“not used; omit if not appearing elsewhere”, “only if used elsewhere”) have been removed.
  • The Abbreviations section has been streamlined and alphabetically organized for clarity and consistency with journal style.

  1. Relocation of sentence (lines 223–225)

Response:
The sentence was relocated to an earlier part of the Introduction, immediately following the paragraph introducing systemic immunity and the abscopal effect, thereby improving logical flow and narrative progression.

  1. Definition of “ablation cavity” (line 498)

Response:
We added a short definition:

“The ablation cavity refers to the central zone of complete coagulative necrosis formed at the treatment site following ablation.”

  1. Clarify reference time points for hours, days, and months (page 12 onward)

Response:
All temporal indicators have been clarified to specify that the reference time points are (e.g., 24–72 hours post-ablation, day 7–10 post-ablation).
In addition, Section 4.4 (“Standardized Assay Timepoints for Cross-Trial Interpretation”) explicitly defines the baseline and subsequent time points.

  1. Section 4.4.1

“Can be omitted and easily incorporated into Table 2.”

Response:
We have followed this advice and merged Section 4.4.1 into Table 2, making the text more concise and avoiding redundancy.

  1. Chapter 5 introductory sentences

“Please add a few introductory sentences before starting the descriptions.”

Response:
We added a short introductory paragraph at the beginning of Chapter 5 to orient the reader, summarizing the transition from preclinical mechanisms to clinical translation and the section's purpose.

  1. References for Chapters 5.2 and 5.2.1

“Please provide references for chapters 5.2 and 5.2.1”

Response:
We have added appropriate references for these subsections, citing key clinical and translational studies.

  1. Shortening the Conclusion

“The Conclusion is lengthy and essentially repeats earlier content.”

Response:
As suggested, the shortened revised conclusion (lines 683–690) succinctly emphasizes the translational significance of ablation-induced in situ vaccination and outlines future research directions.

We are confident that these revisions have strengthened both the readability and scientific rigor of the review.